behaviour/ecology

animal personality, fitness, parity, reproductive success, social bonds, social networks

**Author for correspondence:**
A. J. Carter
e-mail: alecia.carter@ucl.ac.uk

# Higher sociability leads to lower reproductive success in female kangaroos

C. S. Menz[1], A. J. Carter[2], E. C. Best[1], N. J. Freeman[1], R. G. Dwyer[1,3], S. P. Blomberg[1] and A. W. Goldizen[1]

[1]School of Biological Sciences, The University of Queensland, Brisbane, Queensland, Australia
[2]Department of Anthropology, University College London, London, UK
[3]Global Change Ecology Research Group, University of the Sunshine Coast, Maroochydore, Queensland 4556, Australia

AJC, 0000-0001-5550-9312; NJF, 0000-0003-4911-9857

In social mammals, social integration is generally assumed to improve females' reproductive success. Most species demonstrating this relationship exhibit complex forms of social bonds and interactions. However, female eastern grey kangaroos (*Macropus giganteus*) exhibit differentiated social relationships, yet do not appear to cooperate directly. It is unclear what the fitness consequences of such sociability could be in species that do not exhibit obvious forms of cooperation. Using 4 years of life history, spatial and social data from a wild population of approximately 200 individually recognizable female eastern grey kangaroos, we tested whether higher levels of sociability are associated with greater reproductive success. Contrary to expectations, we found that the size of a female's social network, her numbers of preferential associations with other females and her group sizes all negatively influenced her reproductive success. These factors influenced the survival of dependent young that had left the pouch rather than those that were still in the pouch. We also show that primiparous females (first-time breeders) were less likely to have surviving young. Our findings suggest that social bonds are not always beneficial for reproductive success in group-living species, and that female kangaroos may experience trade-offs between successfully rearing young and maintaining affiliative relationships.

## 1. Introduction

Group living, or sociality, is thought to evolve when the net benefits of associating with conspecifics outweigh the costs [1]. Sociality can offer increased protection from predators, increased access to mates and cooperation in finding or defending food resources, while costs include increased intraspecific competition, aggression and disease transmission [2]. Unlike the evolutionary processes leading to

group living, the fitness costs and benefits of *differentiated* social relationships—where individuals associate with differing degrees with different conspecifics—have only recently received attention [3]. We use the term 'sociability' as an overarching term describing individual animals' social patterns, including two quite different components. The first is 'gregariousness', referring to an individual's propensity to be near others regardless of their identity (as measured by, for example, group size or nearest neighbour distance). The second, 'differentiated social relationships' describes an individual's direct and indirect relationships with particular conspecifics and can be quantified using a range of social network metrics and measures of relationship preferences. Differentiated social relationships can arise from individuals' preferences for or avoidances of particular other conspecifics, but may also reflect pairs of individuals spending more or less time together simply because of patterns of home range overlap. Conspecifics vary consistently in their patterns of both gregariousness and differentiated social relationships [4,5] and this variation is known to be heritable in other species [6,7], suggesting that these traits can be a target for selection.

Interest in the fitness benefits of sociability (and in particular differentiated social relationships) has increased rapidly in recent years, led by Silk *et al.* [8] reporting that female yellow baboons (*Papio cynocephalus*) who had closer relationships and were more socially integrated within their group experienced greater infant survival. Other studies have since found similar positive relationships between various dimensions of sociability and reproductive success for females in other mammals [9–16] and for males in birds and mammals [17–19]. Positive relationships between sociability metrics and survival have also been found in a range of mammals [13,20–23]. A number of mechanisms have been proposed as possible explanations for positive links between fitness and the differentiated social relationships component of sociability (reviewed by Thompson [24]). These include reduced stress levels, which can improve immune function [25–27], protection against predators [28] and against harassment by males [9], thermal benefits [29], and reduced vigilance allowing more time for feeding [13,30].

By contrast to the increasing number of studies showing positive effects of sociability on fitness, studies of some other species of mammals have found that social integration can reduce aspects of individuals' fitness, at least in some circumstances. For example, juvenile white-faced capuchin monkeys (*Cebus capucinus*) are vulnerable to infanticide, and the young of the most social (and most dominant) females were more likely to be killed after the replacement of the alpha male compared to the young of less dominant females [15]. In yellow-bellied marmots (*Marmota flaviventris*), females with higher affiliation strengths weaned fewer offspring over a year [31] and had lower overwinter survival [32]. However, more socially integrated yearling females and more gregarious adult females had increased survival over summer in these marmots [33], suggesting that the influences of sociability on survival in marmots are complex.

Why highly sociable individuals experience such contrasting fitness outcomes across species is unclear; however, some of the contrasts between species and across studies may come from differences in the metrics of sociability used across studies [34]. For example, direct metrics describe aspects of sociability such as gregariousness (e.g. mean group sizes), the strengths and longevity of social bonds, or the numbers of an individual's direct connections (e.g. the social network measure 'degree', which is the sum of an individual's association partners; [35]). Indirect metrics represent concepts like the extent to which an individual connects others in its social network (e.g. 'betweenness') or whether the individual's associates are also associates of each other (e.g. 'clustering coefficient') [36]. Indeed, measures of sociability differed in their ability to predict longevity, an aspect of fitness, in semi-free ranging rhesus macaques (*Macaca mulatta*), highlighting the possible effects of different measures of sociability on a study's conclusions [34]. Alternatively, these variable fitness outcomes may arise because social organizations differ among species, in particular, whether species are cooperative and/or display overtly positive interactions. This is because some of the proposed mechanisms for sociability-related fitness benefits require overt cooperation (e.g. protection of females from harassment by males, communal defence against predators) or allogrooming (stress reduction). We could thus predict that greater sociability may increase fitness more in cooperative, social species than in non-cooperative, social species.

The vast majority of studies of the effects of differentiated social relationships on fitness have been conducted on primates and other cooperative, social species, leading to a growing conclusion that these positively affect reproductive success and survival. Studies of social but non-cooperative species are thus critical. We explored the fitness correlates of individual differences in both gregariousness and differentiated social relationships in the eastern grey kangaroo (*Macropus giganteus*; hereafter called 'kangaroo'), a large-bodied grazing macropod that is among the most gregarious of the macropods [37]. As in other species with high fission–fusion social dynamics [38], individuals aggregate in temporary foraging groups that frequently change in size and composition [39,40].

Kangaroos are interesting subjects for studies of sociability and fitness because individuals show consistent differences in measures of their sociability [41] and display differentiated social relationships, including significant preferences and avoidances of conspecifics [30,42] that are only weakly explained by relatedness [42,43]. However, they do not display any apparent cooperative behaviour, or even allogrooming, apart from that between mothers and dependent offspring. This lack of cooperation or overt positive interactions suggests that patterns of sociability may have less influence than food availability and predation risk on successful reproduction in kangaroos. Indeed, King *et al.* [44] found that young kangaroos had higher survival from permanent emergence from the pouch to 21 months of age when their mothers spent more time alone with their young (more than 10 m from conspecifics), suggesting a negative relationship between gregariousness and reproductive success in this species, but the relationships between reproductive success and other dimensions of sociability remain unstudied.

We investigated how individual females' sociability traits related to their reproductive success using 4 years of data on social grouping patterns and reproductive records from a population containing over 200 female kangaroos at any one time. Our aim was to investigate how different measures of sociability influenced whether females raised a young in a given year to (i) the large pouch-young stage, (ii) permanent emergence from the pouch and (iii) weaning. We predicted that any influence of sociability would be strongest during the last stage, when young are no longer carried by their mother and are most vulnerable to predation. Our measures of sociability included both social network analysis (SNA) metrics [36,45] and non-network measures [34]. Specifically, we measured females' network strengths and clustering coefficients, and females' average group sizes. In addition, we included a fourth sociability metric, females' numbers of preferred associates, which is a measure of relationship preferences that controls for range overlap. This last variable integrated both spatial and social data to help identify pairs of individuals that fed together more frequently than expected given their respective patterns of home range use (i.e. 'preferred associates'). This measure differs from 'degree', which quantifies the number of different individuals a particular kangaroo was seen with as part of a feeding group and ignores individuals' space use.

# 2. Methods

## 2.1. Field data collection

Field observations were conducted between January 2010 and December 2013 at Sundown National Park in Queensland, Australia (28°55′03″ S, 151°34′46″ E). The study site spanned 37.4 ha of a patchwork of pasture and eucalypt woodland, bordered on all sides by forest (see [46], for detailed descriptions [47]). Data on monthly rainfall were obtained from the Bureau of Meteorology website for Texas Post Office (station number 041100, 28°51′00″ S, 151°10′12″ E; located approx. 40 km west of the study site). Rainfall totals for the years 2010–2013 varied from 617.9 mm in 2012 to 874.3 mm in 2010; all annual totals exceeded the average yearly total rainfall for the previous 5 years (2005–2009; mean = 614.3 mm yr$^{-1}$ ± s.e. 45.1 mm yr$^{-1}$). There were no predators of adult kangaroos at the study site, as there were no dingos (*Canis lupus dingo*) present, but foxes (*Vulpes vulpes*) and wedge-tailed eagles (*Aquila audax*), which take young kangaroos, were present.

The kangaroos that inhabited the site had been studied since 2009. We estimate that 300 kangaroos grazed regularly within the site, including approximately 240 females in any given year. Individual female kangaroos were identified based on their natural features and markings [48]. We checked the accuracy of identification using a catalogue of photographs, measures of inter-observer reliability and repeated genotyping of individuals that were otherwise difficult to identify [46].

Data on social associations were collected as described by Best *et al.* [46]. Each month between 10 and 16 surveys were conducted within the study site in the 2 h following sunrise or the 2 h prior to sunset. From January 2010 until September 2012, surveys were conducted once a day, alternating between mornings and evenings, when the kangaroos were feeding. Between October 2012 and December 2013, surveys were conducted either once or twice a day (once in the morning and once in the evening). The 15 m chain rule [49] was used to assign group membership; any individual who was within 15 m of at least one other group member was included in the group. Female kangaroos at the study site had been previously observed to maintain social and spatial cohesion at this distance [50]. All members of a group were considered to be associating (called the 'gambit of the group'; [51]). Estimates of group size included all adult and sub-adult females within a group but did not include

males of any age. This was done because we were primarily interested in females' association patterns with each other. Mature adult males frequently moved among groups of females searching for mating opportunities, and sub-adult males most often grouped with other males of similar size and age classes [46]. We recorded the location of the centre of the group (or, where possible, the approximate location of each group member) using a Garmin eTrex H GPS (Garmin International Inc., Olathe, KS, USA). During field sessions in which surveys were not being conducted, GPS locations were taken for females ad libitum from 2010 until mid-2012.

We visually assessed the reproductive state of female kangaroos based on the presence and developmental stage of their young. Kangaroos give birth after 36 days of gestation to a small (approx. 800 mg), blind and hairless young that then continues its development inside the mother's pouch [52]. Young are permanently excluded from the pouch (PEP) at approximately 46 weeks after birth, although they begin to leave the pouch on temporary excursions approximately one to two months prior to PEP [53]. After PEP they continue to nurse for approximately another seven months before they are weaned [54]. The reproductive state of each female we encountered during association surveys was recorded in one of six categories, as described by Best *et al.* [46]: females with no visible pouch-young (NPY), females with a small pouch-young (SPY), females with a medium pouch-young (MPY), females with a large pouch-young (LPY), females with a young-at-foot (YAF) (young that had reached PEP but continued to nurse and remain close to its mother) and females who were caring simultaneously for both a SPY and a young-at-foot (SPY, YAF). For the analyses in this paper, we were interested in the success that females had at rearing young to the LPY, PEP and weaning stages. A young was considered a LPY when it was too large to completely fit in the pouch and started leaving it for small periods. Based on observations of nursing duration following PEP [55], young were considered weaned eight months from their recorded month of PEP unless observed suckling after this date. Female offspring that were independent from their mothers (weaned) but too young to breed were recorded as sub-adults.

## 2.2. Data analysis—general approach

We conducted two sets of models to address our general aim. These differed in the timeframes over which the data were aggregated to accommodate different measures of sociability. Our first approach used a larger dataset to investigate the relationships between a female's success at rearing young to the LPY, PEP and weaning stages and three measures of sociability (average group size, strength and clustering coefficient). We used a second approach in order to be able to add a fourth social variable to the analysis—the number of other females that a given female went out of her way to spend time feeding with (number of preferred associates). To accumulate sufficient home range data on enough females to determine these preferred associations, we needed 2 years of data for the explanatory variables for this second set of models. This reduced our sample sizes, which is why we present both sets of models—Approach 1 with a larger sample but fewer (three) social variables and Approach 2 with the smaller sample but four social variables.

In both approaches, for each female for a given year, we analysed (i) whether she produced a young that survived to the LPY stage, (ii) to PEP, and (iii) to weaning. Macropods have a unique reproductive physiology that allows up to three dependent offspring at one time (one in embryonic diapause, one in the pouch and one still nursing). Thus females are able to raise one offspring to weaning per year even though their offspring are nutritionally dependent for more than 1 year. We included analyses of the LPY and PEP stages to determine which stages of offspring growth are most affected by mothers' sociability metrics. Female parity (whether they had previously bred) was included in all models, based on past reproductive states and/or previous offspring. Individuals of unknown parity status for a given year were excluded from the analysis. All models included female ID as a random effect on the intercept. All analyses were conducted in R (v. 3.3.3) with individuals' social network metrics calculated using the igraph package [56].

## 2.3. Data analysis—Approach 1

This approach used social association data and reproductive records for kangaroos spanning 12-month calendar-year periods and collected between January 2010 and December 2013. Only females of reproductive age that were recorded on association surveys at least 10 times in a year were included in the analyses. We restricted our analysis only to females whose reproductive state had been recorded at least once every two months because of the possibility that we may have missed

reproductive milestones for less frequently seen animals. In our dataset, 126 females met these criteria. This resulted in the inclusion of a total of 285 lines of data in each model ($N = 68$ in 2010, 78 in 2011, 59 in 2012 and 80 in 2013). Some of the females were included in multiple years.

For each year, association survey data were used to generate a matrix of pairwise association indices among all adult females who had been recorded at least 10 times in that year using the half-weight index (HWI) [57]. On average (median), females were recorded 64 times/year (range 11–137; annual median range = 56.5–97 observations/individual/year). We then calculated two social network metrics for each individual (two variables representing aspects of females' patterns of differentiated social relationships): 'strength', the sum of association indices for an individual; and local 'clustering coefficient', a measure of how well connected an individual's neighbours were among themselves [35]. We chose the clustering coefficient as the most relevant indirect network metric for kangaroos, based on the prediction that females might gain further foraging benefits when foraging with others who are familiar to each other. Females have been previously found to be less vigilant when with females with whom they spent more time [30]. Strength and clustering coefficient were not correlated in our sample (Spearman rank correlation: $S = 3\,483\,100$, $\rho = 0.10$, $p = 0.10$) and thus capture different aspects of individuals' social lives.

The median group size in which a female was encountered during surveys in a year was included as a measure of gregariousness, which represents a different aspect of sociability to the social network metrics. However, since this variable was strongly correlated with strength (Spearman correlation test: $S = 1\,753\,500$, $\rho = 0.545$, $p < 0.001$), we used the residuals of a linear-mixed effects model with median group size regressed against strength and female ID as a random effect. Thus, this variable represents an individual's general gregariousness, while controlling for the time spent with particular others.

The reproductive and social data were analysed using three binomial (logit link) generalized linear mixed models (GLMMs) fit by maximum likelihood with a Laplace approximation. These models calculated the probabilities that, for a given year, a female raised a young to at least to the LPY stage (model 1.1), the PEP stage (model 1.2) and to the weaning stage (model 1.3). In all models, the explanatory variables were individuals' strength, clustering coefficient and median annual group size, and the number of sightings of the individual during surveys, parity (primiparous, multiparous) and year. Primiparous females are first-time breeders while multiparous ones have bred previously. The year was included as an explanatory variable to test for effects of different environmental conditions among years.

Colinearity among the explanatory variables was checked by calculating variance inflation factors (VIFs) from the raw data. In all models, the VIF for year ranged between 5.33 and 5.64. However, since annual rainfall is predicted to affect multiple aspects of the kangaroos' behaviour, this was not unexpected, and we included year as a control variable in the models. Some social network metrics are expected to increase with observation frequency in networks where not every individual is observed in each sampling period [58], and indeed this correlation between sample size and social network metrics has been described in another population of eastern grey kangaroos [59]. For the females included in our analyses (individuals with a minimum of 10 observations per year at a frequency of at least once per two months), strength but not clustering coefficient was significantly correlated with the number of sightings (strength: Spearman rank correlation: $S = 1\,847\,800$, $\rho = 0.470$, $p < 0.001$; clustering coefficient: Spearman rank correlation: $S = 39\,87\,700$, $\rho = -0.036$, $p = 0.547$); we thus included observation frequency as an explanatory variable in all models to control for this effect.

Because the relational data underlying social networks are not independent, these data violate the assumptions of 'traditional' inferential statistics and alternative approaches are necessary to determine the statistical significance of the estimated effect [58]. To this end, we: (i) performed a series of permutations of the networks (described below), (ii) re-calculated individuals' network metrics after each permutation, and then (iii) re-ran the above analyses with individuals' metrics from the randomly generated networks (see [60] for details). To determine whether the observed effect was significantly associated with the response (probability of production of young that survived to a given stage), it was compared to the effect sizes estimated from the models using the metrics obtained from each randomized network, permuted under the null hypothesis of random association. The $p$-value for the variable was calculated as the number of times that the observed effect size was larger (or smaller) than the effect from the permuted data. Because social network metrics can be sensitive to differences in group sizes [58], we performed swaps of individuals in the observation-level data, known as data-stream permutations [60,61], keeping the number of groups and the sizes of groups constant. This is in contrast to network-level permutations, where nodes or edges of the network are swapped. We also constrained the individual swaps to occur within observation sessions, which

avoids individuals being swapped into groups that it was not possible for them to be observed in (for example, on a day when they were not observed). Each permutation swapped 100 individuals between groups (for reference, numbers of observations of all individuals ranged between 6148 and 10 107 per year, thus 100 swaps per permutation represents a modest change to the network). Because the change per permutation was relatively modest, to ensure the randomized networks were sufficiently different to the observed network, we discarded the first 200 permutations, then performed a further 1000 permutations from which the individuals' network metrics were recalculated each time.

## 2.4. Data analysis—Approach 2

We also used data collected over 2-year periods to assess the relationships between individuals' sociability and offspring survival, incorporating a measure of females' association preferences, which could not be calculated over single-year periods. Association survey data were grouped into three overlapping 2-year periods (January 2010–December 2011 = time period A, January 2011–December 2012 = time period B, and January 2012–December 2013 = time period C). Data were included only for reproductively mature females who had been recorded on at least 10 association surveys in a given 2-year period, and for which a minimum of 50 GPS location fixes had been recorded during that time. This resulted in sample sizes of 87, 78 and 79 females for time periods A, B and C, respectively (total = 244 lines of data for each model, representing 110 different individuals). Because some females were included in multiple years, female ID was included as a random effect in analyses. The same two social network metrics used in Approach 1, strength and clustering coefficient, were calculated for each of these 244 cases for use as explanatory variables in the analyses.

To include females' numbers of preferred associates as an explanatory variable in the analysis, we calculated which pairwise associations between females were stronger than expected based on the pair's degree of spatial overlap. This was done following the method outlined by Best *et al.* [42], using the DigiRoo2 software package in R [62], which randomizes association survey data based on individuals' space use to create a null model for pairwise associations. For each time period, only females with 50 or more GPS location points were included, as this was the minimum number of points needed to obtain stable home range estimates [42]. From these spatial data, 200 simulated surveys were generated based on parameters from observed surveys; each survey simulated the entire original 2-year window of data, resulting in more than 10 000 simulated observations across more than 250 observation sessions across the 200 simulations. These simulated survey data were entered into SOCPROG2.4 [63] to generate expected association indices for all dyads based on individuals' space use, controlling for individual differences in gregariousness by using a variation on the half-weight index (the HWIG, see [64]) as the association index. If the observed association index was in the top 2.5% of the expected values from the 200 simulations, the pair was categorized as preferred associates. The total number of preferred associates that a female had over a 2-year period, which ranged from zero to 22, was then used as a measure of her sociability.

To test for relationships between females' sociability and whether they produced a young that survived to the LPY stage (model 2.1), the PEP stage (model 2.2) and to weaning (model 2.3), GLMMs fitted by maximum likelihood were performed using explanatory variables from these 2-year datasets. The binomial response variables related to the second year of the 2-year time period. This ensured that the same young were not counted in two time periods (e.g. a young that survived to PEP in 2012 would otherwise have been counted in both periods B and C) and thus there was no pseudoreplication in the response variables. Social explanatory variables included a female's residual median group size (controlling for her strength), number of preferred associates, strength and clustering coefficient scores, all measured over the 2-year time period. Other explanatory variables included parity, the number of times each female had been recorded on surveys, and the time period in question. The explanatory variables were checked for colinearity prior to running the models, and all variables had VIFs below 5.

## 3. Results

We present only the statistically significant results here; results for non-significant variables are presented in the tables. For ease of reading, we have put all statistical values in the tables, without repeating them in the text.

**Table 1.** Model results from generalized linear mixed models explaining the variation in whether females produced young that survived to each listed developmental state, for single calendar years (Approach 1). Numbers in bold represent $p < 0.05$. Note that $p$-values for the social network metrics were generated through permutations and thus a $z$-value is not reported.

| response | fixed effects | estimate | s.e. | $z$-value | $p$-value |
|---|---|---|---|---|---|
| Model 1.1: probability of producing a young that survived to LPY ($n = 285$ observations, 126 females) | intercept | 2.455 | 1.613 | 1.522 | 0.128 |
| | strength | −0.296 | 0.230 | NA | 0.212 |
| | clustering coefficient | −2.520 | 2.080 | NA | 0.254 |
| | year-2011[a] | −0.362 | 0.509 | −0.712 | 0.477 |
| | year-2012[a] | 0.569 | 0.500 | 1.138 | 0.255 |
| | year-2013[a] | 0.105 | 0.521 | 0.202 | 0.840 |
| | **parity**[b] | **−1.970** | **0.518** | **−3.806** | **<0.001** |
| | residual group size | −0.051 | 0.170 | −0.302 | 0.763 |
| | **number of sightings** | **0.02** | **0.010** | **2.107** | **0.035** |
| Model 1.2: probability of producing a young that survived to PEP ($n = 285$ observations, 126 females) | intercept | 1.342 | 1.351 | 0.993 | 0.321 |
| | **strength** | **−0.524** | **0.192** | **NA** | **0.000** |
| | clustering coefficient | −0.827 | 1.765 | NA | 0.510 |
| | year-2011[a] | −0.272 | 0.435 | −0.625 | 0.532 |
| | year-2012[a] | 0.476 | 0.421 | 1.131 | 0.258 |
| | year-2013[a] | −0.365 | 0.442 | −0.826 | 0.409 |
| | **parity**[b] | **−1.375** | **0.48** | **−2.864** | **0.004** |
| | residual group size | −0.099 | 0.145 | −0.682 | 0.495 |
| | **number of sightings** | **0.016** | **0.008** | **2.076** | **0.038** |
| Model 1.3: probability of producing a young that survived to weaning ($n = 282$ observations, 124 females) | intercept | −1.535 | 1.545 | −0.994 | 0.32 |
| | **strength** | **−0.880** | **0.238** | **NA** | **0.000** |
| | clustering coefficient | 0.335 | 2.035 | NA | 0.698 |
| | **year-2011**[a] | **1.045** | **0.481** | **2.17** | **0.030** |
| | **year-2012**[a] | **2.459** | **0.525** | **4.681** | **<0.001** |
| | **year-2013**[a] | **1.164** | **0.51** | **2.28** | **0.023** |
| | **parity**[b] | **−3.758** | **1.084** | **−3.467** | **0.001** |
| | residual group size | −0.054 | 0.168 | −0.318 | 0.750 |
| | **number of sightings** | **0.036** | **0.01** | **3.717** | **<0.001** |

[a]Reference category: year-2010.
[b]Reference category: multiparity.

## 3.1. Approach 1

Over a 12-month period, primiparous females were significantly less likely to produce offspring that survived to each of the three stages (LPY, PEP, weaning) than did multiparous females (table 1). Females with high strength had a lower probability of raising a young to PEP or weaning than those with lower scores on this metric (table 1 and figure 1). Females were significantly more likely to raise a young to weaning in 2011, 2012 and 2013 than in 2010 (table 1). There was a significant positive relationship between the number of sightings of a female and her likelihood of rearing a young to any stage.

## 3.2. Approach 2

Primiparous mothers were less likely to produce an offspring that survived to weaning than were multiparous females (table 2). Similarly, females with a larger number of preferred associates, greater strength and larger group sizes for a given strength were less likely to wean a young than females

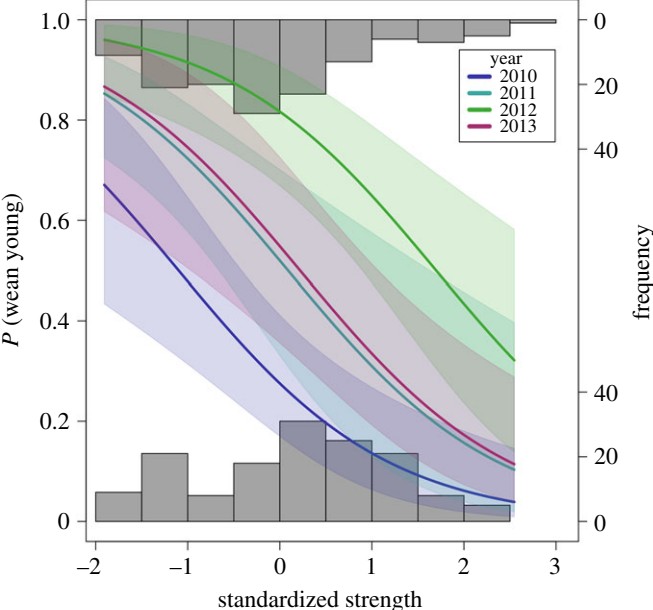

**Figure 1.** The effects of female sociability on the probability of weaning a young for Approach 1. Predicted probabilities (solid lines) and 95% confidence intervals (shading) of a multiparous female kangaroo successfully weaning a young in each of the 4 years from 2010 to 2013 against standardized strength scores. Frequencies of observed weaning outcomes (0 = failure, 1 = success) for individual females are shown in the histograms.

with fewer preferred associates, lower strength and smaller group sizes for a given strength (table 2 and figure 2). In addition, females were less likely to wean young in time period B than in time period A. There were small but positive relationships between the numbers of sightings of females and their likelihoods of having young survive to LPY and weaning (table 2).

## 4. Discussion

Many recent studies of group-living mammals have reported fitness benefits of sociability, and particularly of differentiated social relationships, in females ([9,10,63], reviewed by [64]). We analysed a multi-year dataset to explore the social factors influencing offspring production, a key component of lifetime fitness, in wild eastern grey kangaroos. By contrast to most studies on other mammals, we found robust negative relationships between multiple dimensions of females' sociability and their reproductive success, while controlling for other factors including a female's parity. These findings confirm those of King et al. [44] for another population of eastern grey kangaroos, who showed that YAF had higher survival if their mothers spent more time away from other conspecifics. By using a combination of SNA techniques and analyses of social preferences, we were able to extend this earlier finding to show that female kangaroos with (i) larger social networks, (ii) greater numbers of preferred associates (females with whom they spent more time than expected) and (iii) larger median group sizes produced fewer young that survived to weaning. There was no effect of sociability on numbers of young surviving to the LPY or PEP stages, indicating that more sociable mothers had lower survival of their young due to mortality between PEP and weaning. Below we briefly discuss effects on survival of young of our three non-social explanatory variables (parity, year and the number of sightings of individuals), then possible mechanisms that might explain the negative effects of sociability that we found, before discussing hypotheses proposed to explain both positive and negative sociability-fitness relationships found in other species, and the ways in which kangaroos differ from those species. Finally, we discuss the wider implications of our findings for understanding the link between sociability and fitness.

Year, parity and the number of sightings of females all related to their chances of raising offspring in our analyses. Parity had a significant effect on offspring survival in four of our six models. Reproductive success increases with age in many species of mammal, as more experienced females can better protect their young or target their maternal care more effectively [65]. Primiparous females in our study were aged between 29 and 54 months when they first bred, and the youngest females in this category may

**Table 2.** Model results from generalized linear mixed models explaining the variation in females' production of young that survived to each listed developmental state, for overlapping 2-year time periods (Approach 2). Numbers in bold represent $p <$ 0.05. Note that $p$-values for the social network metrics were generated through permutations and thus a $z$-value is not reported.

| response | fixed effect | estimate | s.e. | $z$-value | $p$-value |
|---|---|---|---|---|---|
| Model 2.1: probability of producing a young that survived to LPY ($n = 241$ observations, 110 females) | intercept | 4.406 | 3.38 | 1.304 | 0.192 |
| | strength | −0.444 | 0.383 | NA | 0.368 |
| | clustering coefficient | −5.882 | 4.096 | NA | 0.05 |
| | number of preferred associates | 0.031 | 0.047 | 0.647 | 0.518 |
| | time period B[a] | 0.35 | 0.446 | 0.785 | 0.433 |
| | time period C[a] | 0.61 | 0.797 | 0.765 | 0.444 |
| | parity[b] | −0.842 | 0.598 | −1.408 | 0.159 |
| | residual group size | −0.295 | 0.327 | −0.902 | 0.367 |
| | **number of sightings** | **0.017** | **0.008** | **2.01** | **0.044** |
| Model 2.2: probability of producing a young that survived to PEP ($n = 241$ observations, 110 females) | **intercept** | **5.193** | **2.495** | **2.081** | **0.038** |
| | strength | −0.539 | 0.265 | NA | 0.485 |
| | clustering coefficient | −6.191 | 2.994 | NA | 0.225 |
| | number of preferred associates | 0.002 | 0.03 | 0.071 | 0.944 |
| | time period B[a] | −0.047 | 0.343 | −0.138 | 0.891 |
| | time period C[a] | 0.262 | 0.571 | 0.459 | 0.646 |
| | parity[b] | −0.839 | 0.448 | −1.874 | 0.061 |
| | residual group size | −0.132 | 0.254 | −0.52 | 0.603 |
| | number of sightings | 0.009 | 0.005 | 1.722 | 0.085 |
| Model 2.3: probability of producing a young that survived to weaning ($n = 235$ observations, 109 females) | intercept | 1.64 | 2.728 | 0.601 | 0.548 |
| | **strength** | **−1.251** | **0.324** | **NA** | **0.009** |
| | clustering coefficient | −0.969 | 3.261 | NA | 0.309 |
| | **number of preferred associates** | **−0.093** | **0.037** | **−2.528** | **0.011** |
| | **time period B[a]** | **−0.916** | **0.406** | **−2.258** | **0.024** |
| | time period C[a] | 0.482 | 0.658 | 0.733 | 0.463 |
| | **parity[b]** | **−2.285** | **0.712** | **−3.208** | **0.001** |
| | **residual group size** | **−0.719** | **0.32** | **−2.252** | **0.024** |
| | **number of sightings** | **0.027** | **0.007** | **3.904** | **<0.001** |

[a]Reference category: time period A.
[b]Reference category: multiparity.

not have reached full adult body size by this time [66]. Such small primiparous mothers may have been more stressed by the energetic demands of carrying pouch-young than were older mothers. We included year as an explanatory variable, rather than a random effect, to understand which stages of offspring growth might be most affected by environmental factors. Differences among years emerged only for survival from PEP to weaning, not for younger offspring, suggesting that factors such as food availability, kangaroo density and/or the number of predators in the area influence the survival of young most during their vulnerable period prior to weaning. Further research is needed to better understand these effects. Our finding that females who were sighted more often were more likely to have surviving young requires further research to understand. We do not believe that less gregarious females were less likely to be seen on our surveys, as all kangaroos were easy to note in the usually short grass of our paddocks. The finding could be explained by our having missed seeing some young

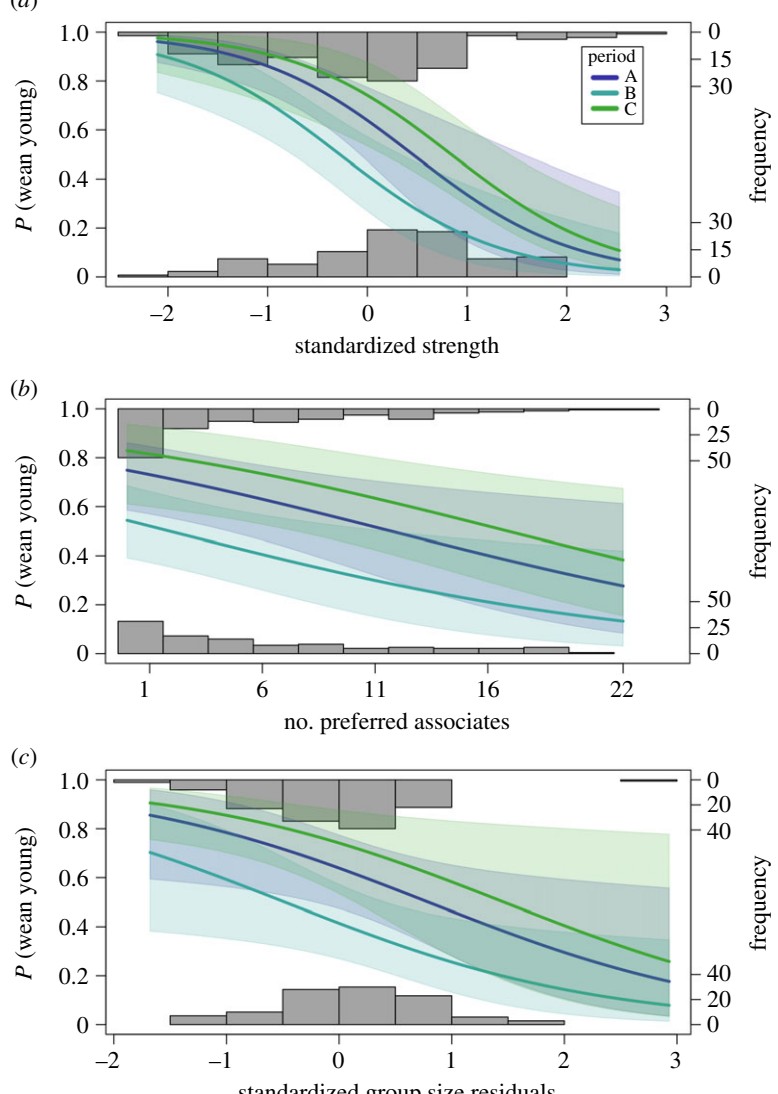

**Figure 2.** The effects of female sociability on the probability of weaning a young for Approach 2. Predicted probabilities (solid lines) and 95% confidence intervals (shading) of a multiparous female weaning a young in the second year of a 2-year period (period A = 2010–2011, period B = 2011–2012, period C = 2012–2013) against (a) standardized strength scores, (b) number of preferred associates, (c) standardized group size residuals. Frequencies of observed weaning outcomes (0 = failure, 1 = success) for individual females are shown in the histograms.

for females seen less frequently, but another possibility is that females seen more often spent more time in the open paddocks at the study site and as a result had higher food intakes than females who spent more of their time in the surrounding woodland. While the significance of this variable is interesting, it has been controlled for in the analyses, which were focused on the effects of the social variables.

To understand why higher levels of sociability might be negatively related to female kangaroos' production of young, we first consider the other mammal in which negative sociability-fitness relationships have been best shown, the yellow-bellied marmot. This marmot is a large and facultatively social ground squirrel [67], with a different social structure to that of kangaroos. Females can live alone, with a single male and their offspring or in multi-female matrilineal groups. Closely related females do allogroom and may cooperatively thermoregulate sometimes, but otherwise are not cooperative [68]. Negative effects of sociability on the fitness of females have been shown in this species. Female marmots who were more sociable during summer had lower annual reproductive success [31] and lower survival during their subsequent winter hibernation; these females also had reduced longevity [67]. However, they were more likely to survive over the summer [33]. These studies suggest a trade-off in relation to sociability between a female's survival and that of her offspring, but the mechanisms that explain this negative link are not yet completely clear. Adult

females who were in larger groups were more likely to survive the summer [33], perhaps due to a lower predation risk, while the lower reproductive success of more sociable females may have been due to density effects [31].

The marmot studies may provide some insight into our findings. The negative effect of group size on the production of young to weaning in kangaroos could be a result of local competition for food, as is hypothesized to be the case for the marmots, and was also suggested for the observation of lower reproductive success in more gregarious female red-necked wallabies [69]. However, the negative relationships between fitness and strength and between fitness and a female's number of preferential relationships in our kangaroos require other explanations. Considering the three phases of offspring development in kangaroos separately could provide a key. The probability of raising an offspring to weaning was the most strongly and negatively related to all of our measures of a mother's sociability. Strength was also significantly negatively related to the annual production of young to PEP, but this effect was not as strong. Together, these results suggest that the critical period is between permanent emergence and weaning, when young are nutritionally dependent on their mother but unable to be transported in her pouch. Two non-mutually exclusive explanations are possible. First, there may be an increased likelihood of the young of more sociable mothers becoming separated from their mother [44]. Separation would keep a young from its mother's milk and increase its chance of being taken by a predator. This could occur both when groups fission normally, which may be more frequent for highly sociable females and especially those with more preferred associates, with whom they go out of their way to feed. Separation is also more likely to occur during a disturbance when multiple individuals panic and flee. YAF occasionally follow females that are not their mothers ([70]; C. Menz, N. Freeman 2010–2013 and A. Carter 2006, personal observation). In larger groups, there is a greater chance that a young might mistake another individual for its mother and become separated from its mother. Observations of 'adoptions' in kangaroos suggest that mother-young recognition may not be well developed [70]. Second, the time and cognitive effort needed by females with more differentiated social relationships to keep track of interactions with others could also reduce those mothers' time available for activities such as maintaining contact with young and anti-predator vigilance, as suggested by Montero *et al.* [33] for yellow-bellied marmots and Sabol *et al.* [71] for prairie voles (*Microtus ochrogaster*).

It is also interesting to consider how kangaroos might differ from the mammals for which sociability has been found to positively relate to females' fitness. Ostner & Schülke [72] review five categories of mechanisms that might explain the relationship between sociability and fitness in primates, on which most of this research has been done. These include (i) agonistic coalitions affecting dominance ranks, (ii) individuals protecting each other from harassment, (iii) cooperative defence of resources, (iv) thermoregulation benefits from huddling, and (v) reductions in predation risk for more sociable females. The first four of these are unlikely to apply to female kangaroos, and many other mammals. There is no evidence that kangaroos form agonistic coalitions, support each other against harassment, defend resources or huddle for thermoregulation. Allogrooming also does not occur in kangaroos except between mothers and their dependent young, and females do not assist each other with offspring care. The absence of any clear cooperative behaviours probably explains the lack of positive effects of sociability on offspring production in kangaroos. There is some evidence linking sociability to antipredator behaviour in kangaroos. For instance, some studies have found that vigilance decreases with group size, suggesting that more gregarious females may be at lower risk of predation, or at least gain more feeding time (e.g. [49,50]). Carter *et al.* [30] found that females were less vigilant and fed more when they had a higher association index with their nearest neighbour. Despite these possible links between sociability and antipredator behaviour in kangaroos, these links did not increase the reproductive success of more sociable females, as we found a negative relationship between sociability and offspring production.

A positive relationship between the differentiated social relationships dimension of sociability and fitness has also been found for two graminivorous mammal species, big-horned sheep (*Ovis canadensis*) and feral horses (*Equus ferus caballus*). Feral horses live in stable bands containing groups of unrelated females and either a single or occasionally multiple adult males [73]. Females' social integration scores (including measures of proximity, approaches and grooming) were significantly positively related to their birth rates and less strongly to the survival of those foals, and negatively related to the rate at which females were harassed by both males and other females [9,73]. Harassment reduces females' reproductive success, so the fitness benefit to sociability in this species appears to be caused by social integration, which reduces harassment. While female kangaroos do experience harassment from males, this is not at the level experienced by horses, and aggression is rare among female kangaroos [40], suggesting that harassment is not serious enough to influence the fitness of female kangaroos. Female big-horned sheep have a similar fission–fusion social system to that of kangaroos, although average

rates of pairwise association are higher in the sheep [13]. As with kangaroos, relatedness plays only a limited role in the association patterns of female big-horned sheep, there is no cooperative defence against predators or of resources and females do not assist each other in raising offspring. However, unlike the kangaroos in our study, Vander Wal et al. [13] found that both strength and eigenvector centrality (a measure of an individual's connectedness to others that incorporates both direct and indirect connections in the network) positively predicted both the production of female lambs and their own survival but not the survival of their lambs. The mechanisms promoting higher survival and lamb production for more sociable females are as yet unclear; thus we cannot yet understand why big-horned sheep and kangaroos show opposite patterns in relation to sociability and fitness.

Given the negative relationship between sociability and fitness we have found, further research is required to understand why kangaroos and many other mammals display differentiated social relationships, and why the propensity and ability to form these relationships have initially evolved in mammals. It is possible that more sociable female kangaroos have higher survival themselves, at least in locations that have non-negligible predation rates from dingos, due both to safety in numbers and the benefits of being able to better interpret the alarm behaviours of close associates (e.g. [28,30]). This could resemble the apparent fitness trade-offs of being more sociable that are seen for female yellow-bellied marmots; high sociability could increase a female's survival and thus her future reproduction while reducing her current reproductive success. It is also possible that the costs and benefits of sociability to female kangaroos vary with environmental conditions (such as food availability, population density and predation risk) and may differ between populations. For example, in some mammals, group size is correlated positively with fitness benefits when food is abundant but negatively when conditions are poor (reviewed by Ebensperger et al. [74]). Montero et al. [33] suggested that it is more costly for female marmots to maintain social relationships when predation risk is higher, due to the incompatibility of paying attention to conspecifics and watching for predators. What is clear is that we need further studies of the relationship between sociability and fitness in a broader range of species, to counter the current bias towards studies of primates.

Ethics. This research was approved by the University of Queensland Animal Ethics Committee (AEC approval nos. SIB/206/09/(NF) and SIB/142/12/ARC) and permission was granted by the Queensland Parks and Wildlife Service to conduct research for scientific purposes at Sundown National Park (permit nos. WITK06125709 and WITK11362312).

Data accessibility. The data for this manuscript have been deposited in the Dryad Digital Repository: https://dx.doi.org/10.5061/dryad.79cnp5hsb [75].

Authors' contributions. A.W.G., E.C.B and C.S.M. conceived the ideas and designed methodology; C.S.M., E.C.B. and N.J.F. collected the data; C.S.M., S.P.B., R.G.D. and A.J.C. analysed the data; C.S.M., A.J.C. and A.W.G. led the writing of the manuscript. All authors contributed critically to drafts and gave final approval for publication.

Competing interests. We have no competing interests.

Funding. This work was funded by an Australian Research Council Discovery Grant (grant no. DP120102693) to A.W.G., and made possible by an Australian Postgraduate Award scholarship to C.S.M. and a Northcote Graduate Scholarship to E.C.B.

Acknowledgements. We thank Peter Hasselgrove and Ian Elms, rangers with the Queensland Department of National Parks, Sport and Racing, for their support for fieldwork for this project.

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
