## [Reviewer comments · Royal Society Open Science]

Review History

RSOS-200950.R0 (Original submission)

Review form: Reviewer 1

Is the manuscript scientifically sound in its present form?

Yes

Are the interpretations and conclusions justified by the results?

Yes

Is the language acceptable?

Yes

Do you have any ethical concerns with this paper?

No

Have you any concerns about statistical analyses in this paper?

Yes

Recommendation?

Major revision is needed (please make suggestions in comments)

Comments to the Author(s)

General comments:

The authors address the relationships between sociability and reproductive success in female kangaroos, and find that the more sociable females (as measured by node strength, clustering coefficient, group size, and number of preferred associates) have reduced reproductive success.

Overall, the manuscript well-written. The abstract, the introduction, and the discussion all have a clear logical structure with a good flow. I have a few questions about the methods and the results as outlined below and in the more detailed comments that follow. I hope the authors will find these useful for improving their manuscript.

1) I see that the first analysis is focused on each year, while the second analysis includes two-year periods. Is that the only difference between them? If so, based on the information given in the manuscript, it is not clear why both of these approaches (data analysis approach 1 & 2) are essential. In addition, why is the "preferred associates" analysis not included in approach 1?

2) There is some overlap between the periods used in analysis 2, as also acknowledged by the authors. For instance, one period covers January 2010-December 2011, while another period covers January 2011-December 2012. The overlap between these periods raises a concern for pseudoreplication, and I recommend that the authors address this issue as well as provide a detailed explanation for the choice of these time periods.

3) The authors used randomization to deal with the lack of independence in the network data. Yet, as far as I know, the randomization technique used in this study is a rather uncommon one. Has it been used in other studies before? I strongly recommend adding more information and more details about this technique, as it is difficult to fully understand its suitability based on the given information. The details can be added either in the main text or in the supplementary materials.

4) Are data available to analyze whether the identity of the females' preferred associates are stable through time and reproductive stages? An exciting addition to the manuscript, if the data are available, is whether females associate with the same conspecifics regardless of their reproductive stage, or if the identity of their preferred associate changes through different stages. For example, do they display homophily and associate with conspecifics of the same reproductive state?

Detailed comments:

Lines 19-21: What is meant by social preferences? Preference to be with others? Preference to associate with specific social partners? The abstract is the only place the term "social preferences" is used, but it needs to be defined as it will mean different things to different researchers. I would actually recommend replacing it with a different term that is used more frequently in the text.

Lines 25-29: Any ideas for why? Is the reduced reproductive success linked to increased resource competition (e.g. females who are around other conspecifics will have to share or compete for resources) or aggression from others (e.g. being around other conspecifics increases the chance that the offspring will encounter aggression). Or another reason?

Line 30: Please define "primiparous".

Lines 46-49: It is good that this distinction is clarified early in the manuscript. I suggest adding a note that gregariousness is one aspect of sociability (as mentioned on lines 83-85).

Lines 136-138: I see the note about controlling for range overlap, but it is not clear if/how this measure is different from degree. Please clarify.

Lines 173-174: Was females' age known in more detail than adult or sub-adult? If so, can it be added to models?

Lines 190-196: These six categories are nice to know about, but it is not clear how they were utilized during the data analysis, given that the analysis focuses only on LPY, PEP, and weaning. Please explain this more in the methods (and results if applicable).

Lines 204-354: See my general comments for questions about the data analysis approach used in this study.

Lines 205-207: More information is needed here about the differences between the two approaches, and the reasons for why both are needed for this manuscript.

Line 228: Minor point, but please write the number (i.e. 126 instead of one hundred and twenty six).

Lines 229-230: What does "female-year observations" mean? Also, clarify if the numbers in parenthesis represent the number of individuals or the number of observations.

Line 247: Why "further"? Isn't "group size" the only measure of gregariousness in this manuscript?

Line 253: Where is the fourth measure (number of preferred associates)?

Line 261: Please define "primiparous" and "multiparous". Primiparous isn't defined until line 362.

Line 304: What does "a burn-in of permutations" mean?

Line 319: What is "female/time periods"? This term and "female-year observations" need to be clearly defined.

Lines 366-367: Do authors have any idea on why there is a difference between the years? Is it due to ecological or social factors? Would be good to address in the discussion.

Lines 367-369: This is an interesting finding, and I see a similar pattern also exists for the second set of analyses (lines 394-397). I recommend talking about this pattern in the discussion to address if it might be a byproduct of data collection or if it actually has a biological meaning. Authors could address some questions such as: What factors influence the likelihood of sighting an individual? Do data collection methods bias sightings towards certain individuals? Are individuals in groups sighted more than others? What are some potential biological reasons for why frequently sighted individuals are more likely to rear young successfully? Do the frequently sighted individuals forage more (or in better locations) and thus able to nurse young better?

Line 392: Following up on my earlier comment on lines 136-138, it is still not clear to me if "females with a large number of preferred associates" differs from "females with a high degree". It is fine to stick with "preferred associates" but it will help the readers to know whether or not this measure is the same as degree.

Lines 457-463: The first thought I had when reading the abstract was that the lower reproduction success of social females could be due to competition for resources or aggression from others, so I was glad to see that the authors addressed these possibilities and suggested alternative explanations.

Line 513: I appreciate and agree with the point that more research on non-primates is needed to understand the relationships between reproductive success and sociability. However, I think this message is emphasized in some places where it is not necessary to do so. For instance, “non-primate” is not necessary here on line 513. Same is true for line 496, because the four explanations also do not apply to many taxa besides mammals. Removing these will not distract from authors’ message that more research on other species is needed.

Lines 560-568: Although this paragraph makes an important point, it is not a strong conclusion paragraph and looks almost as if it is an afterthought. I recommend moving it to an earlier part of discussion and ending with the previous paragraph (lines 539-558).

Review form: Reviewer 2

Is the manuscript scientifically sound in its present form?

Yes

Are the interpretations and conclusions justified by the results?

No

Is the language acceptable?

Yes

Do you have any ethical concerns with this paper?

No

Have you any concerns about statistical analyses in this paper?

No

Recommendation?

Accept with minor revision (please list in comments)

Comments to the Author(s)

Dear Editor and Authors,

This study examines the relationship between spatial association and individual fitness in a population of eastern grey kangaroos. Overall, the manuscript is very well-written. The introduction summarizes the relevant literature to nicely frame the research question, and the data collection methods and analytical approach are clearly and comprehensively described, and subsequently well-executed. Unusually, for me, I have very few specific comments, as this information is so clearly and thoroughly presented.

I do, however, have a conceptual bugbear that underpins the theoretical framework of the manuscript and interpretation of results, that I believe should be addressed.

Throughout the manuscript, there appears to be some confusion surrounding the interchangeability of the terms ‘gregariousness’ and ‘sociality’ (or ‘sociability’, for the individual). As the authors correctly point out (L46), these are conceptually different phenomena. Gregariousness refers to the spatial association observed in group-living animals, whereas sociability refers to “differentiated social relationships” typically underpinned by social exchange. The authors appear to delineate these two concepts in their species under the guise of selective proximity regulation toward certain individuals (L48), but I am not convinced that this alone transcends gregariousness in to the realms of sociability. The premise of sociability is that

individuals selectively associate with one another. These sociable associations, as described by Seyfarth & Cheney [3], and many others, are more than proximity regulation, but also tend to involve affiliative exchange; e.g., grooming, preening, fin-rubbing. Although the authors acknowledge themselves that gregariousness and sociable are conceptually different, the cross-talking between these two phenomena has resulted in some confusion in the logic of the text, and, I would argue, inaccuracies in the interpretations of the data collected and analyzed. Importantly, I must add, I certainly do not believe this detracts from the value of these findings, but I do believe the paper would be significantly improved if these conceptual arguments were more precisely navigated.

For example, to support the view that sociability can be costly to individual fitness (L73-80), the authors cite two studies, one of which looks at sociability in yellow-bellied marmots, and another that looks at gregariousness in red-necked wallabies. This, as well as reflecting a confusion of concepts, also highlights my point above. In the marmot studies [32,33], the authors defined sociability as the exchange of affiliative behaviors such as greetings and allogrooming. In the wallaby study [31], and another on grey kangaroos [the current study species: 45], the authors labeled the spatial associations between individuals as gregariousness; and reserve the term sociability to the affiliative exchange between mother and offspring. Given the definitions described here, as well as, to my knowledge, the literature as a whole, the spatial associations analyzed in the current manuscript seem to represent inter-individual differences in gregariousness of the grey kangaroos, and not sociability.

The authors state that, unlike the earlier study on grey kangaroos by King et al. [45], the current study examines other dimensions of sociability. While this is analytically true, in the sense that the current authors use social network analyses to handle their data, the variables that goes in to these analyses are the same spatial associations. To put it somewhat crudely, the use of social network analyses does not make the association data going in to the model more social than it is. That is not to say that sociability cannot have negative fitness consequences (see the marmots), but the association data presented in the current manuscript I believe represents gregariousness, and not sociability.

In conclusion, therefore, I do not believe that the current study is tackling a question of sociability and its fitness consequences, but rather gregariousness (albeit selective in terms of membership) and its fitness consequences. As such, the finding in the current study that close-associations have a negative impact on fitness (as observed by King et al.), can perhaps be more parsimoniously explained, without the need to do so in the context of the costs and benefits of sociability and cooperation etc. Instead, as achieved by King, the findings presented in the current manuscript could also be explained by the positive effect of maternal care, and mother-infant isolation, on offspring survival.

Minor analytical comments:

L266: Given 'Year' is not the focus of your analyses (you define it as a control: L268), might it be better to include this within your random effect? This may well improve you model, and would avoid the rather clumsy interpretation of 2011 vs 2012 vs 2013 etc.

L225: With a benchmark of ten observation/subject, is there a cause for concern for how robust these models are? Is there some sort of 'power-analysis' that underpins whether a network model is robust enough to make predictions about network structure?

Overall, I really enjoyed reading this manuscript. I believe this study is of great value and would be of interest to the readers of your journal. I do suggest, however, that it would benefit from a re-framing under the 'fitness costs of (selective) gregariousness', and not the 'fitness costs of sociability'; which tends to assume more than inter-individual proximity regulation.

Decision letter (RSOS-200950.R0)

Dear Dr Carter,

The editors assigned to your paper ("Higher sociability leads to lower reproductive success in female kangaroos") have now received comments from reviewers. We would like you to revise your paper in accordance with the referee and Associate Editor suggestions which can be found below (not including confidential reports to the Editor). Please note this decision does not guarantee eventual acceptance.

Please submit a copy of your revised paper before 29-Jul-2020. Please note that the revision deadline will expire at 00.00am on this date. If we do not hear from you within this time then it will be assumed that the paper has been withdrawn. In exceptional circumstances, extensions may be possible if agreed with the Editorial Office in advance. We do not allow multiple rounds of revision so we urge you to make every effort to fully address all of the comments at this stage. If deemed necessary by the Editors, your manuscript will be sent back to one or more of the original reviewers for assessment. If the original reviewers are not available, we may invite new reviewers.

- Data accessibility

If you wish to submit your supporting data or code to Dryad (<http://datadryad.org/>), or modify your current submission to dryad, please use the following link:
<http://datadryad.org/submit?journalID=RSOS&manu=RSOS-200950>

- Competing interests

- Authors' contributions

- Acknowledgements

- Funding statement

on behalf of Dr Kimberley Mathot (Associate Editor) and Kevin Padian (Subject Editor)
openscience@royalsociety.org

Associate Editor's comments (Dr Kimberley Mathot):

Two reviewers and I have examined the manuscript which investigates the relationship between spatial associations and fitness in a population of eastern grey kangaroos. We all found the manuscript to be very well-written, with a strong methodological framework and analytical approach. There are nonetheless a few areas that the referees have highlighted where the text could be clarified. These should all be possible for the authors to address, and I am therefore recommending major revisions.

Specifically, reviewer #1 raises some questions about the distinction between analytical approach 1 and 2. They differ in the number of years included, and the binning (1 versus 2 years). I agree that the specific contrast between the approaches is not immediately clear. Adding one or two

sentences after lines 205-206 summarizing the key differences would be helpful. Referee #1 also raises concerns about pseudo replication in analysis 2, and provides interesting suggestions for follow-up work (e.g., stability of preferred associates through time and across reproductive stages). While such analyses may be beyond the scope of the current paper, you might speculate on them in the discussion.

Reviewer #2 points out confusion around the interchangeability of the terms 'gregariousness' and 'sociality/sociability'. Specifically, reviewer #2 points out that the data used in the analyses is about spatial associations rather than the specific details of the type of interaction. In this way, it is more appropriately labelled "gregariousness". Based on the references provided by reviewer #2, I agree, and would request that the authors rephrase throughout to clarify that their study addresses gregariousness and fitness consequences, and not sociability and fitness consequences. I recognize that their measures of spatial affiliation are richer than simple gregariousness measures as they also take into account the specific individuals with whom the focal is spatially affiliated, and this may be worth discussing.

Comments to Author:

Reviewers' Comments to Author:

Reviewer: 1

Comments to the Author(s)

General comments:

The authors address the relationships between sociability and reproductive success in female kangaroos, and find that the more sociable females (as measured by node strength, clustering coefficient, group size, and number of preferred associates) have reduced reproductive success.

Overall, the manuscript well-written. The abstract, the introduction, and the discussion all have a clear logical structure with a good flow. I have a few questions about the methods and the results as outlined below and in the more detailed comments that follow. I hope the authors will find these useful for improving their manuscript.

1) I see that the first analysis is focused on each year, while the second analysis includes two-year periods. Is that the only difference between them? If so, based on the information given in the manuscript, it is not clear why both of these approaches (data analysis approach 1 & 2) are essential. In addition, why is the "preferred associates" analysis not included in approach 1?

2) There is some overlap between the periods used in analysis 2, as also acknowledged by the authors. For instance, one period covers January 2010-December 2011, while another period covers January 2011-December 2012. The overlap between these periods raises a concern for pseudoreplication, and I recommend that the authors address this issue as well as provide a detailed explanation for the choice of these time periods.

3) The authors used randomization to deal with the lack of independence in the network data. Yet, as far as I know, the randomization technique used in this study is a rather uncommon one. Has it been used in other studies before? I strongly recommend adding more information and more details about this technique, as it is difficult to fully understand its suitability based on the given information. The details can be added either in the main text or in the supplementary materials.

4) Are data available to analyze whether the identity of the females' preferred associates are stable through time and reproductive stages? An exciting addition to the manuscript, if the data are available, is whether females associate with the same conspecifics regardless of their reproductive stage, or if the identity of their preferred associate changes through different stages. For example, do they display homophily and associate with conspecifics of the same reproductive state?

Detailed comments:

Lines 19-21: What is meant by social preferences? Preference to be with others? Preference to associate with specific social partners? The abstract is the only place the term “social preferences” is used, but it needs to be defined as it will mean different things to different researchers. I would actually recommend replacing it with a different term that is used more frequently in the text.

Lines 25-29: Any ideas for why? Is the reduced reproductive success linked to increased resource competition (e.g. females who are around other conspecifics will have to share or compete for resources) or aggression from others (e.g. being around other conspecifics increases the chance that the offspring will encounter aggression). Or another reason?

Line 30: Please define “primiparous”.

Lines 46-49: It is good that this distinction is clarified early in the manuscript. I suggest adding a note that gregariousness is one aspect of sociability (as mentioned on lines 83-85).

Lines 136-138: I see the note about controlling for range overlap, but it is not clear if/how this measure is different from degree. Please clarify.

Lines 173-174: Was females’ age known in more detail than adult or sub-adult? If so, can it be added to models?

Lines 190-196: These six categories are nice to know about, but it is not clear how they were utilized during the data analysis, given that the analysis focuses only on LPY, PEP, and weaning. Please explain this more in the methods (and results if applicable).

Lines 204-354: See my general comments for questions about the data analysis approach used in this study.

Lines 205-207: More information is needed here about the differences between the two approaches, and the reasons for why both are needed for this manuscript.

Line 228: Minor point, but please write the number (i.e. 126 instead of one hundred and twenty six).

Lines 229-230: What does “female-year observations” mean? Also, clarify if the numbers in parenthesis represent the number of individuals or the number of observations.

Line 247: Why “further”? Isn’t “group size” the only measure of gregariousness in this manuscript?

Line 253: Where is the fourth measure (number of preferred associates)?

Line 261: Please define “primiparous” and “multiparous”. Primiparous isn’t defined until line 362.

Line 304: What does “a burn-in of permutations” mean?

Line 319: What is “female/time periods”? This term and “female-year observations” need to be clearly defined.

Lines 366-367: Do authors have any idea on why there is a difference between the years? Is it due to ecological or social factors? Would be good to address in the discussion.

Lines 367-369: This is an interesting finding, and I see a similar pattern also exists for the second set of analyses (lines 394-397). I recommend talking about this pattern in the discussion to address if it might be a byproduct of data collection or if it actually has a biological meaning. Authors could address some questions such as: What factors influence the likelihood of sighting an individual? Do data collection methods bias sightings towards certain individuals? Are individuals in groups sighted more than others? What are some potential biological reasons for why frequently sighted individuals are more likely to rear young successfully? Do the frequently sighted individuals forage more (or in better locations) and thus able to nurse young better?

Line 392: Following up on my earlier comment on lines 136-138, it is still not clear to me if “females with a large number of preferred associates” differs from “females with a high degree”. It is fine to stick with “preferred associates” but it will help the readers to know whether or not this measure is the same as degree.

Lines 457-463: The first thought I had when reading the abstract was that the lower reproduction success of social females could be due to competition for resources or aggression from others, so I was glad to see that the authors addressed these possibilities and suggested alternative explanations.

Line 513: I appreciate and agree with the point that more research on non-primates is needed to understand the relationships between reproductive success and sociability. However, I think this message is emphasized in some places where it is not necessary to do so. For instance, “non-primate” is not necessary here on line 513. Same is true for line 496, because the four explanations also do not apply to many taxa besides mammals. Removing these will not distract from authors’ message that more research on other species is needed.

Lines 560-568: Although this paragraph makes an important point, it is not a strong conclusion paragraph and looks almost as if it is an afterthought. I recommend moving it to an earlier part of discussion and ending with the previous paragraph (lines 539-558).

Reviewer: 2

Comments to the Author(s)
Dear Editor and Authors,

This study examines the relationship between spatial association and individual fitness in a population of eastern grey kangaroos. Overall, the manuscript is very well-written. The introduction summarizes the relevant literature to nicely frame the research question, and the data collection methods and analytical approach are clearly and comprehensively described, and subsequently well-executed. Unusually, for me, I have very few specific comments, as this information is so clearly and thoroughly presented.

I do, however, have a conceptual bugbear that underpins the theoretical framework of the manuscript and interpretation of results, that I believe should be addressed.

Throughout the manuscript, there appears to be some confusion surrounding the interchangeability of the terms ‘gregariousness’ and ‘sociability’ (or ‘sociability’, for the individual). As the authors correctly point out (L46), these are conceptually different phenomena. Gregariousness refers to the spatial association observed in group-living animals, whereas sociability refers to “differentiated social relationships” typically underpinned by social exchange. The authors appear to delineate these two concepts in their species under the guise of selective proximity regulation toward certain individuals (L48), but I am not convinced that this alone transcends gregariousness in to the realms of sociability. The premise of sociability is that individuals selectively associate with one another. These sociable associations, as described by Seyfarth & Cheney [3], and many others, are more than proximity regulation, but also tend to

involve affiliative exchange; e.g., grooming, preening, fin-rubbing. Although the authors acknowledge themselves that gregariousness and sociable are conceptually different, the cross-talking between these two phenomena has resulted in some confusion in the logic of the text, and, I would argue, inaccuracies in the interpretations of the data collected and analyzed. Importantly, I must add, I certainly do not believe this detracts from the value of these findings, but I do believe the paper would be significantly improved if these conceptual arguments were more precisely navigated.

For example, to support the view that sociability can be costly to individual fitness (L73-80), the authors cite two studies, one of which looks at sociability in yellow-bellied marmots, and another that looks at gregariousness in red-necked wallabies. This, as well as reflecting a confusion of concepts, also highlights my point above. In the marmot studies [32,33], the authors defined sociability as the exchange of affiliative behaviors such as greetings and allogrooming. In the wallaby study [31], and another on grey kangaroos [the current study species: 45], the authors labeled the spatial associations between individuals as gregariousness; and reserve the term sociability to the affiliative exchange between mother and offspring. Given the definitions described here, as well as, to my knowledge, the literature as a whole, the spatial associations analyzed in the current manuscript seem to represent inter-individual differences in gregariousness of the grey kangaroos, and not sociability.

The authors state that, unlike the earlier study on grey kangaroos by King et al. [45], the current study examines other dimensions of sociability. While this is analytically true, in the sense that the current authors use social network analyses to handle their data, the variables that goes in to these analyses are the same spatial associations. To put it somewhat crudely, the use of social network analyses does not make the association data going in to the model more social than it is. That is not to say that sociability cannot have negative fitness consequences (see the marmots), but the association data presented in the current manuscript I believe represents gregariousness, and not sociability.

In conclusion, therefore, I do not believe that the current study is tackling a question of sociability and its fitness consequences, but rather gregariousness (albeit selective in terms of membership) and its fitness consequences. As such, the finding in the current study that close-associations have a negative impact on fitness (as observed by King et al.), can perhaps be more parsimoniously explained, without the need to do so in the context of the costs and benefits of sociability and cooperation etc. Instead, as achieved by King, the findings presented in the current manuscript could also be explained by the positive effect of maternal care, and mother-infant isolation, on offspring survival.

Minor analytical comments:

L266: Given 'Year' is not the focus of your analyses (you define it as a control: L268), might it be better to include this within your random effect? This may well improve you model, and would avoid the rather clumsy interpretation of 2011 vs 2012 vs 2013 etc.

L225: With a benchmark of ten observation/subject, is there a cause for concern for how robust these models are? Is there some sort of 'power-analysis' that underpins whether a network model is robust enough to make predictions about network structure?

Overall, I really enjoyed reading this manuscript. I believe this study is of great value and would be of interest to the readers of your journal. I do suggest, however, that it would benefit from a re-framing under the 'fitness costs of (selective) gregariousness', and not the 'fitness costs of sociability'; which tends to assume more than inter-individual proximity regulation.

Author's Response to Decision Letter for (RSOS-200950.R0)

See Appendix A.

Decision letter (RSOS-200950.R1)

Dear Dr Carter,

It is a pleasure to accept your manuscript entitled "Higher sociability leads to lower reproductive success in female kangaroos" in its current form for publication in Royal Society Open Science.

Please ensure that you send to the editorial office the individual files for each table included in your manuscript. You can send these in a zip folder if more convenient. Failure to provide these files may delay the processing of your proof. You may disregard this request if you have already provided these files to the editorial office.

on behalf of Dr Kimberley Mathot (Associate Editor) and Kevin Padian (Subject Editor)
openscience@royalsociety.org

Associate Editor Comments to Author (Dr Kimberley Mathot):

Associate Editor

Comments to the Author:

Thank you for your clear and thoughtful responses to the comments/suggestions you received in the first round of review. I am satisfied that all the major comments have been addressed or rebutted as appropriate, and am recommending the article be accepted.

Appendix A

Dear Dr Mathot,

RE: Higher sociability leads to lower reproductive success in female kangaroos

Responses to reviewers' comments

We have now revised our manuscript in line with the reviewers' comments. We believe our revised manuscript is now clearer, thanks to the constructive feedback we received. In brief, our major changes have: (1) clarified the main differences between our analytical approaches, and (2) used clearer definitions of sociability, gregariousness, and differentiated social relationships in non-human animals. We provide our detailed responses below in **red text** after each comment.

We thank the editor for the opportunity to resubmit this manuscript.

Best wishes,

Dr Alecia Carter, on behalf of the co-authors

Associate Editor's comments (Dr Kimberley Mathot):

Two reviewers and I have examined the manuscript which investigates the relationship between spatial associations and fitness in a population of eastern grey kangaroos. We all found the manuscript to be very well-written, with a strong methodological framework and analytical approach. There are nonetheless a few areas that the referees have highlighted where the text could be clarified. These should all be possible for the authors to address, and I am therefore recommending major revisions.

Specifically, reviewer #1 raises some questions about the distinction between analytical approach 1 and 2. They differ in the number of years included, and the binning (1 versus 2 years). I agree that the specific contrast between the approaches is not immediately clear. Adding one or two sentences after lines 205-206 summarizing the key differences would be helpful. Referee #1 also raises concerns about pseudo replication in analysis 2, and provides interesting suggestions for follow-up work (e.g., stability of preferred associates through time and across reproductive stages). While such analyses may be beyond the scope of the current paper, you might speculate on them in the discussion.

Reviewer #2 points out confusion around the interchangeability of the terms 'gregariousness' and 'sociality/sociability'. Specifically, reviewer #2 points out that the data used in the analyses is about spatial associations rather than the specific details of the type of interaction. In this way, it is more appropriately labelled "gregariousness". Based on the references provided by reviewer #2, I agree, and would request that the authors rephrase throughout to clarify that their study addresses gregariousness and fitness consequences, and not sociability and fitness consequences. I recognize that their measures of spatial affiliation are richer than simple gregariousness measures as they also take into account the specific individuals with whom the focal is spatially affiliated, and this may be worth discussing.

We thank the editor for these encouraging comments. We have now addressed these main points, which are detailed in our responses below.

Comments to Author:

Reviewers' Comments to Author:

Reviewer: 1

Comments to the Author(s)

General comments:

The authors address the relationships between sociability and reproductive success in female kangaroos, and find that the more sociable females (as measured by node strength, clustering coefficient, group size, and number of preferred associates) have reduced reproductive success.

Overall, the manuscript well-written. The abstract, the introduction, and the discussion all have a clear logical structure with a good flow. I have a few questions about the methods and the results as outlined below and in the more detailed comments that follow. I hope the authors will find these useful for improving their manuscript.

We thank the reviewer for these positive and helpful comments.

1) I see that the first analysis is focused on each year, while the second analysis includes two-year periods. Is that the only difference between them? If so, based on the information given in the manuscript, it is not clear why both of these approaches (data analysis approach 1 & 2) are essential. In addition, why is the “preferred associates” analysis not included in approach 1?

See our response to comment relating to lines 205-207 below.

2) There is some overlap between the periods used in analysis 2, as also acknowledged by the authors. For instance, one period covers January 2010–December 2011, while another period covers January 2011–December 2012. The overlap between these periods raises a concern for pseudoreplication, and I recommend that the authors address this issue as well as provide a detailed explanation for the choice of these time periods.

We understand the concern about the overlap between the periods, but this only relates to the explanatory variables. There is no overlap in the response variables, as for each two-year period, we have used the females’ reproductive success data from the second year only (thus 2011, 2012 and 2013). We have justified the need for two-year periods for the calculation of preferred associates elsewhere (lines 222-229). We have added the following to clarify the lack of pseudoreplication: “This ensured that the same young were not counted in two time periods (e.g. a young that survived to PEP in 2012 would otherwise have been counted in both periods B and C) and thus there was no pseudoreplication in the response variables.” (lines 379-380)

Regarding the explanatory variables, it is possible that the overlap in periods could lead to lower variation, which would, if anything, underestimate the effect sizes. Pseudoreplication should be a cause of concern if it artificially inflated the sample size, but this is not the case here, as we were conservative by dropping one year (of the four years) of reproductive success data, and as a result this analysis has less power. Indeed, we find that the results of the first and second set of models parallel each other, but the effect is shallower in the second set of models, in line with the limitations of our conservative approach, and the reduction in variation that the overlapping time periods may result in.

3) The authors used randomization to deal with the lack of independence in the network data. Yet, as far as I know, the randomization technique used in this study is a rather uncommon one. Has it been used in other studies before? I strongly recommend adding more information and more details about this technique, as it is difficult to fully understand its suitability based on the given

information. The details can be added either in the main text or in the supplementary materials.

The randomisation approach for hypothesis testing in social networks was originally recommended by Bejder et al. (1998) and more recently by Farine & Whitehead (2015) and Farine (2017, outlined in Figure 1). The particular data-stream permutations we chose is an approach implemented in a popular animal social network R package, *asnipe* (Farine 2013). Although this approach may not be commonly implemented (Farine (2017 p. 1309) admits that the use of null models “in hypothesis testing is still not widespread”), it is the accepted procedure to deal with non-independence of network data (in theory, if not widely in practice). We have provided further references in this section and terminology to clarify the approach (lines 312-324), but since it has been well-described in several methodological papers, and we feel our description is clear, we have not provided more detail in the text.

Bejder, L., Fletcher, D., & Bräger, S. (1998). A method for testing association patterns of social animals. *Animal behaviour*, 56(3), 719-725.

Farine, D. R. (2013). Animal social network inference and permutations for ecologists in R using *asnipe*. *Methods in Ecology and Evolution*, 4(12), 1187-1194.

Farine, D. R., & Whitehead, H. (2015). Constructing, conducting and interpreting animal social network analysis. *Journal of Animal Ecology*, 84(5), 1144-1163.

Farine, D. R. (2017). A guide to null models for animal social network analysis. *Methods in Ecology and Evolution*, 8(10), 1309-1320.

4) Are data available to analyze whether the identity of the females' preferred associates are stable through time and reproductive stages? An exciting addition to the manuscript, if the data are available, is whether females associate with the same conspecifics regardless of their reproductive stage, or if the identity of their preferred associate changes through different stages. For example, do they display homophily and associate with conspecifics of the same reproductive state?

There are indeed many interesting questions that we could ask about females' preferred associations, including those mentioned by the reviewer, and further questions about whether the stability of relationships relates to females' reproductive success, as has been found for baboons. However, these are beyond the scope of this paper.

Detailed comments:

Lines 19-21: What is meant by social preferences? Preference to be with others? Preference to associate with specific social partners? The abstract is the only place the term “social preferences” is used, but it needs to be defined as it will mean different things to different researchers. I would actually recommend replacing it with a different term that is used more frequently in the text.

We have replaced “social preferences” here with “differentiated social relationships”, which we use in the rest of the manuscript. (line 18)

Lines 25-29: Any ideas for why? Is the reduced reproductive success linked to increased resource competition (e.g. females who are around other conspecifics will have to share or compete for resources) or aggression from others (e.g. being around other conspecifics increases the chance that the offspring will encounter aggression). Or another reason?

These ideas are discussed in the Discussion (lines 502-531), but as the Abstract is already at the 200-word limit and it would take quite a few words to do as suggested here, we have not done this, but could do so if it would be okay to exceed the word limit.

Line 30: Please define “primiparous”.

Done – we have added “first-time breeders”. (Line 28)

Lines 46-49: It is good that this distinction is clarified early in the manuscript. I suggest adding a note that gregariousness is one aspect of sociability (as mentioned on lines 83-85).

We have redefined our terms here (and accordingly throughout the manuscript). “We use the term ‘sociability’ as an overarching term describing individual animals’ social patterns, including two quite different components. The first is ‘gregariousness’, referring to an individual’s propensity to be near others regardless of their identity (as measured by, for example, group size or nearest neighbour distance). The second, ‘differentiated social relationships’, describes an individual’s direct and indirect relationships with particular conspecifics and can be quantified using a range of social network metrics and measures of relationship preferences. Differentiated social relationships can arise from individuals’ preferences for or avoidances of particular other conspecifics, but may also reflect pairs of individuals spending more or less time together simply because of patterns of home range overlap. Conspecifics vary consistently in their patterns of both gregariousness and differentiated social relationships [4,5] and this variation is known to be heritable in other species [6,7], suggesting that these traits can be a target for selection.” (lines 45-55)

Lines 136-138: I see the note about controlling for range overlap, but it is not clear if/how this measure is different from degree. Please clarify.

We have added the following here to clarify this. “This last variable integrated both spatial and social data to help identify pairs of individuals that fed together more frequently than expected given their respective patterns of home range use (i.e. ‘preferred associates’). This measure differs from ‘degree’, which quantifies the number of different individuals a particular kangaroo was seen with as part of a feeding group and ignores individuals’ space use.” (lines 144-149)

Lines 173-174: Was females’ age known in more detail than adult or sub-adult? If so, can it be added to models?

It would have been great to be able to add Age as a variable in our models. However, while we knew the ages of females who became adults during our study, many of the females were already adults when we began the study and thus we didn’t know their exact ages.

Lines 190-196: These six categories are nice to know about, but it is not clear how they were utilized during the data analysis, given that the analysis focuses only on LPY, PEP, and weaning. Please explain this more in the methods (and results if applicable).

We have added the following after these lines to explain which of these stages are important in our analyses: “For the analyses in this paper, we were interested in the success that females had at rearing young to the LPY, PEP and weaning stages.” (lines 208-209)

We would like to leave the explanation of the six stages in the manuscript to allow readers a full understanding of macropods’ reproduction.

Lines 204-354: See my general comments for questions about the data analysis approach used in this study.

We have addressed those general comments above.

Lines 205-207: More information is needed here about the differences between the two approaches, and the reasons for why both are needed for this manuscript.

We have added the following here to address this comment. “Our first approach used a larger dataset to investigate the relationships between a female’s success at rearing young to the LPY, PEP and weaning stages and three measures of sociability (average group size, strength and clustering coefficient). We used a second approach in order to be able to add a fourth social variable to the analysis – the number of other females that a given female went out of her way to spend time feeding with (# of preferred associates). To accumulate sufficient home range data on enough females to determine these preferred associations, we needed two years of data for the explanatory variables for this second set of models. This reduced our sample sizes, which is why we present both sets of models – Approach 1 with a larger sample but fewer (three) social variables and Approach 2 with the smaller sample but four social variables.” (lines 219-229)

We also added the following at the start of the description of the Approach 2 analyses to remind readers why we needed that second approach: “We also used data collected over two-year periods to assess the relationships between individuals’ sociability and offspring survival, incorporating a measure of females’ association preferences, which could not be calculated over single-year periods.” (lines 338-340)

Line 228: Minor point, but please write the number (i.e. 126 instead of one hundred and twenty six).

This was written out because it is grammatically incorrect to start a sentence with a numeral. To address this, we have reworded the sentence to move the numeral: “In our dataset, 126 females met these criteria.” (lines 252-253)

Lines 229-230: What does “female-year observations” mean? Also, clarify if the numbers in parenthesis represent the number of individuals or the number of observations.

We have changed the wording here to the following: “This resulted in the inclusion of a total of 285 lines of data in each model (N = 68 in 2010, 78 in 2011, 59 in 2012 and 80 in 2013). Some of the females were included in multiple years.” (lines 253-255)

Line 247: Why “further”? Isn’t “group size” the only measure of gregariousness in this manuscript?

We have removed “further” from this sentence.

Line 253: Where is the fourth measure (number of preferred associates)?

This measure is not included in the models used in Approach 1, which is why it is not mentioned here. It is mentioned in the description of Approach 2.

Line 261: Please define “primiparous” and “multiparous”. Primiparous isn’t defined until line 362.

We have added the following here and removed the definition that had been at line 362.

“Primiparous females are first-time breeders while multiparous ones have bred previously.” (lines 290-291)

Line 304: What does “a burn-in of permutations” mean?

This term is frequently used during randomisation procedures to describe the practice of discarding initial permutations because these initial permutations are ‘too close’ to the original data. To clarify, we have changed this to “we discarded the first 200 permutations” after the explanation (line 334).

Line 319: What is “female/time periods”? This term and “female-year observations” need to be clearly defined.

We have dropped the term “female/time periods” and replaced this sentence with the following: “This resulted in sample of sizes of 87, 78, and 79 females for time periods A, B, and C, respectively (total = 244 lines of data for each model, representing 110 different individuals). Because some females were included in multiple years, female ID was included as a random effect in analyses.” (lines 346-351)

Lines 366-367: Do authors have any idea on why there is a difference between the years? Is it due to ecological or social factors? Would be good to address in the discussion.

We have added the following to the Discussion: “We included year as an explanatory variable, rather than a random effect, to understand which stages of offspring growth might be most affected by environmental factors. Differences among years emerged only for survival from PEP to weaning, not for younger offspring, suggesting that factors such as food availability, kangaroo density and/or the number of predators in the area influence the survival of young most during their vulnerable period prior to weaning. Further research is needed to better understand these effects.” (lines 466-472)

Lines 367-369: This is an interesting finding, and I see a similar pattern also exists for the second set of analyses (lines 394-397). I recommend talking about this pattern in the discussion to address if it might be a byproduct of data collection or if it actually has a biological meaning. Authors could address some questions such as: What factors influence the likelihood of sighting an individual? Do data collection methods bias sightings towards certain individuals? Are individuals in groups sighted more than others? What are some potential biological reasons for why frequently sighted individuals are more likely to rear young successfully? Do the frequently sighted individuals forage more (or in better locations) and thus able to nurse young better?

Detailed analysis of this finding is beyond the scope of this paper, but we have added the following in the Discussion: “Our finding that females who were sighted more often were more likely to have surviving young requires further research to understand. We do not believe that less gregarious females were less likely to be seen on our surveys, as all kangaroos were easy to notice in the usually short grass of our paddocks. The finding could be explained by our having missed seeing some young for females seen less frequently, but another possibility is that females seen more often spent more time in the open paddocks at the study site and as a result had higher food intakes than females who spent more of their time in the surrounding woodland. While the significance of this variable is interesting, it has been controlled for in the analyses, which were focused on the effects of the social variables.” (lines 472-482)

Line 392: Following up on my earlier comment on lines 136-138, it is still not clear to me if “females with a large number of preferred associates” differs from “females with a high degree”. It is fine to stick with “preferred associates” but it will help the readers to know whether or not this measure is the same as degree.

See the response above to the comment relating to lines 136-138. We have now clarified there why these two are very different measures.

Lines 457-463: The first thought I had when reading the abstract was that the lower reproduction success of social females could be due to competition for resources or aggression from others, so I was glad to see that the authors addressed these possibilities and suggested alternative explanations.

Line 513: I appreciate and agree with the point that more research on non-primates is needed to understand the relationships between reproductive success and sociability. However, I think this message is emphasized in some places where it is not necessary to do so. For instance, “non-primate” is not necessary here on line 513. Same is true for line 496, because the four explanations also do not apply to many taxa besides mammals. Removing these will not distract from authors’ message that more research on other species is needed.

We have removed “non-primate” where it was originally.

Lines 560-568: Although this paragraph makes an important point, it is not a strong conclusion paragraph and looks almost as if it is an afterthought. I recommend moving it to an earlier part of discussion and ending with the previous paragraph (lines 539-558).

We have now moved this paragraph to earlier in the Discussion, and added discussion of the effects of Year and Number of sightings to that paragraph. (lines 458-452)

Reviewer: 2

Comments to the Author(s)

Dear Editor and Authors,

This study examines the relationship between spatial association and individual fitness in a population of eastern grey kangaroos. Overall, the manuscript is very well-written. The introduction summarizes the relevant literature to nicely frame the research question, and the data collection methods and analytical approach are clearly and comprehensively described, and subsequently well-executed. Unusually, for me, I have very few specific comments, as this information is so clearly and thoroughly presented.

I do, however, have a conceptual bugbear that underpins the theoretical framework of the manuscript and interpretation of results, that I believe should be addressed.

Throughout the manuscript, there appears to be some confusion surrounding the interchangeability of the terms ‘gregariousness’ and ‘sociality’ (or ‘sociability’, for the individual). As the authors correctly point out (L46), these are conceptually different phenomena. Gregariousness refers to the spatial association observed in group-living animals, whereas sociability refers to “differentiated social relationships” typically underpinned by social exchange. The authors appear to delineate these two concepts in their species under the guise of selective proximity regulation toward certain individuals (L48), but I am not convinced that this alone transcends gregariousness in to the realms of sociability. The premise of sociability is that individuals selectively associate with one another. These sociable associations, as described by Seyfarth & Cheney [3], and many others, are more than proximity regulation, but also tend to involve affiliative exchange; e.g., grooming, preening, fin-rubbing. Although the authors acknowledge themselves that gregariousness and sociable are

conceptually different, the cross-talking between these two phenomena has resulted in some confusion in the logic of the text, and, I would argue, inaccuracies in the interpretations of the data collected and analyzed. Importantly, I must add, I certainly do not believe this detracts from the value of these findings, but I do believe the paper would be significantly improved if these conceptual arguments were more precisely navigated.

We agree that we could have been clearer in the use of our terminology. To this end, we have revised our use of the terminology, and specifically 'sociability', 'gregariousness' and 'differentiated social relationships' and have carefully defined how we use these terms. However, we respectfully disagree with the reviewer's argument here that sociability requires more than 'proximity regulation', needing also overtly positive interactions such as grooming, preening, and fin-rubbing, for at least three reasons. In the first case, such an assumption is species-centric; we cannot assume that a mutual allowance of close proximity between two individuals is less meaningful a demonstration of sociability to a given species than e.g. mutual contact in that species. In the second case, we do demonstrate that individuals 'selectively associate with one another', in line with the reviewer's understanding of sociability. Finally, in species with overtly positive interactions, spatial association and interaction networks often do not correlate (Lehmann & Ross 2011, Castles *et al.* 2014) even though they are often lumped together (e.g. the composite sociality index: Silk *et al.* 2003, 2010). Importantly, both spatial and affiliation networks could be biologically meaningful to a given species in different ways. Furthermore, many studies of cetaceans and birds, for example, measure sociability based on the tendency of individuals to be found in the same group, and do not include measures of 'social exchange' (e.g. Lusseau 2003; Aplin *et al.* 2013). To address this issue, we now differentiate between measures that could reflect only 'gregariousness' (e.g. group size) and measures that reflect 'sociability' (i.e. preferred associates).

Aplin, L. M., Farine, D. R., Morand-Ferron, J., Cole, E. F., Cockburn, A., & Sheldon, B. C. (2013). Individual personalities predict social behaviour in wild networks of great tits (*Parus major*). *Ecology letters*, 16(11), 1365-1372.

Castles, M., Heinsohn, R., Marshall, H. H., Lee, A. E., Cowlshaw, G., & Carter, A. J. (2014). Social networks created with different techniques are not comparable. *Animal Behaviour*, 96, 59-67.

Lehmann, J., & Ross, C. (2011). Baboon (*Papio anubis*) social complexity—a network approach. *American Journal of Primatology*, 73(8), 775-789.

Lusseau, D. (2003). The emergent properties of a dolphin social network. *Proceedings of the Royal Society of London. Series B: Biological Sciences*, 270(suppl_2), S186-S188.

Silk, J. B., Alberts, S. C., & Altmann, J. (2003). Social bonds of female baboons enhance infant survival. *Science*, 302(5648), 1231-1234.

Silk, J. B., Beehner, J. C., Bergman, T. J., Crockford, C., Engh, A. L., Moscovice, L. R., ... & Cheney, D. L. (2010). Strong and consistent social bonds enhance the longevity of female baboons. *Current biology*, 20(15), 1359-1361.

For example, to support the view that sociability can be costly to individual fitness (L73-80), the authors cite two studies, one of which looks at sociability in yellow-bellied marmots, and another that looks at gregariousness in red-necked wallabies. This, as well as reflecting a confusion of concepts, also highlights my point above. In the marmot studies [32,33], the authors defined sociability as the exchange of affiliative behaviors such as greetings and allogrooming. In the wallaby study [31], and another on grey kangaroos [the current study species: 45], the authors labeled the spatial associations between individuals as gregariousness; and reserve the term sociability to the affiliative exchange between mother and offspring. Given the definitions described here, as well as, to my knowledge, the literature as a whole, the spatial associations analyzed in the current manuscript seem to represent inter-individual differences in gregariousness of the grey kangaroos, and not sociability.

We argue that our data are different to those of the wallaby and kangaroo studies that the reviewer mentions here. Those studies measured association indices among individuals (how often they fed in groups together). As we have mentioned in a comment above, two individuals could be seen in groups together either because they go out of their way to be together (our 'preferred associations') or simply because they have overlapping home ranges and thus often feed in the same places. Indeed, some individuals may like to feed in groups, but not care who those group members are. The wallaby and kangaroo studies mentioned by the reviewer did not test for association preferences, so were only able to quantify individuals' gregariousness. We have gone further with our analyses of association preferences.

In relation to the comment above about the marmot study, the marmots do indeed exchange obvious affiliative behaviours, as do primates, so that it makes sense to include these in measures of relationship preferences for those species. The fact that adult kangaroos do not groom each other does not mean they cannot have association preferences, or that we cannot measure these.

The authors state that, unlike the earlier study on grey kangaroos by King et al. [45], the current study examines other dimensions of sociability. While this is analytically true, in the sense that the current authors use social network analyses to handle their data, the variables that goes in to these analyses are the same spatial associations. To put it somewhat crudely, the use of social network analyses does not make the association data going in to the model more social than it is. That is not to say that sociability cannot have negative fitness consequences (see the marmots), but the association data presented in the current manuscript I believe represents gregariousness, and not sociability.

We hope the two comments above have now addressed this concern.

In conclusion, therefore, I do not believe that the current study is tackling a question of sociability and its fitness consequences, but rather gregariousness (albeit selective in terms of membership) and its fitness consequences. As such, the finding in the current study that close-associations have a negative impact on fitness (as observed by King et al.), can perhaps be more parsimoniously explained, without the need to do so in the context of the costs and benefits of sociability and cooperation etc. Instead, as achieved by King, the findings presented in the current manuscript could also be explained by the positive effect of maternal care, and mother-infant isolation, on offspring survival.

As we have argued above, we believe that our results do go beyond what King et al. showed, by showing negative consequences of preferential relationships.

Minor analytical comments:

L266: Given 'Year' is not the focus of your analyses (you define it as a control: L268), might it be better to include this within your random effect? This may well improve you model, and would avoid the rather clumsy interpretation of 2011 vs 2012 vs 2013 etc.

As mentioned above, we have now added the following to the Discussion, explaining why we included Year as a fixed rather than a random effect: "We included year as an explanatory variable, rather than a random effect, to understand which stages of offspring growth might be most affected by environmental factors. Differences among years emerged only for survival from PEP to weaning, not for younger offspring, suggesting that factors such as food availability, kangaroo density and/or perhaps the number of predators in the area influence the survival of young most during their

vulnerable period prior to weaning. Further research is needed to better understand these effects.”
(lines 466-472)

L225: With a benchmark of ten observation/subject, is there a cause for concern for how robust these models are? Is there some sort of ‘power-analysis’ that underpins whether a network model is robust enough to make predictions about network structure?

Our choice to use 10 observations per individual as a minimum threshold was made for two reasons: (1) this number is sufficient to detect biologically-meaningful community network structure in this population (Best et al. 2013), and (2) to balance having sufficient observations per individual with having data on as many individuals as possible. Our networks used a median of 64 observations/individual/year (range 11-137; annual median range = 56.5-97 observations/individual/year), so we tended to have many observations per individual. We now specify this in the manuscript. (line 259-261)

Best, E. C., Seddon, J. M., Dwyer, R. G., & Goldizen, A. W. (2013). Social preference influences female community structure in a population of wild eastern grey kangaroos. *Animal Behaviour*, 86(5), 1031-1040.

Overall, I really enjoyed reading this manuscript. I believe this study is of great value and would be of interest to the readers of your journal. I do suggest, however, that it would benefit from a re-framing under the ‘fitness costs of (selective) gregariousness’, and not the ‘fitness costs of sociability’; which tends to assume more than inter-individual proximity regulation.

We thank the reviewer for the encouraging remarks and detailed comments.